# Dynamical Analyses Show That Professional Archers Exhibit Tighter, Finer and More Fluid Dynamical Control Than Neophytes

**DOI:** 10.3390/e25101414

**Published:** 2023-10-04

**Authors:** Hesam Azadjou, Michalina Błażkiewicz, Andrew Erwin, Francisco J. Valero-Cuevas

**Affiliations:** 1Alfred E. Mann Department of Biomedical Engineering, University of Southern California, Los Angeles, CA 90089, USA; azadjou@usc.edu (H.A.); erwina@usc.edu (A.E.); 2AWF · Department of Physiotherapy, Józef Piłsudski University of Physical Education in Warsaw, 00-968 Warsaw, Poland; michalinablazkiewicz@gmail.com; 3Division of Biokinesiology and Physical Therapy, University of Southern California, Los Angeles, CA 90033, USA

**Keywords:** archery, athletics, human movement, dynamic systems theory, phase space reconstruction, Hurst exponent analysis, sample entropy

## Abstract

Quantifying the dynamical features of discrete tasks is essential to understanding athletic performance for many sports that are not repetitive or cyclical. We compared three dynamical features of the (i) bow hand, (ii) drawing hand, and (iii) center of mass during a single bow-draw movement between professional and neophyte archers: dispersion (convex hull volume of their phase portraits), persistence (tendency to continue a trend as per Hurst exponents), and regularity (sample entropy). Although differences in the two groups are expected due to their differences in skill, our results demonstrate we can *quantify* these differences. The center of mass of professional athletes exhibits tighter movements compared to neophyte archers (6.3 < 11.2 convex hull volume), which are nevertheless less persistent (0.82 < 0.86 Hurst exponent) and less regular (0.035 > 0.025 sample entropy). In particular, the movements of the bow hand and center of mass differed more between groups in Hurst exponent analysis, and the drawing hand and center of mass were more different in sample entropy analysis. This suggests tighter neuromuscular control over the more fluid dynamics of the movement that exhibits more active corrections that are more individualized. Our work, therefore, provides proof of principle of how well-established dynamical analysis techniques can be used to quantify the nature and features of neuromuscular expertise for discrete movements in elite athletes.

## 1. Introduction

Dynamical systems theory has provided multiple tools and techniques to investigate athletic performance. By dynamics, we mean the time-varying interactions between the person and environment while executing a task to different levels of performance [1]. Three useful methods to quantify dynamical performance are its ‘phase portrait’ (a geometric description of how system variables interact with each other over time), the ‘persistence’ of the dynamics (the tendency to continue a current kinematic trend), and the ‘irregularity’ of the dynamics (how fluid the dynamical control is).

Although dynamical techniques are common in applied mathematics and in biological time series [2] and biomechanics [3,4], they have had more limited use in sports as they are applied mostly to cyclical behavior like walking and running [5,6,7,8]. Here, we extend the application of nonlinear dynamics analysis to the domain of discrete movements, focusing on the intricate bow-draw movement.

Archery is an exemplary subject for our investigation due to its unique blend of physicality, mental focus, and historical significance as the bow and arrow could be among the earliest examples of complex projectile weaponry [9,10] and shooting arrows demand a greater level of advanced executive functions within the brain when compared to tasks involving spear-throwing [10]. The archer’s quest for mastery transcends the boundaries of mere physical performance; it delves deep into the realms of cognition and motor control [11,12], and it demands athletes to gain focus and consistency in their movement [13].

In addition, prior studies of archery focus on electromyographic signals [14,15,16,17], posture and stability [18,19], reaction times [20], or effects of eye dominance [21]. However, they do not take a truly dynamical approach.

Here, we specifically study the dynamical features of bow-draw motions to quantify how professional archers attain improved performance. By comparing the dynamics of performance between professional and neophyte archers in the context of the bow-draw movement, we aim to study the differences in motor control strategies and provide a proof of principle for the application of nonlinear dynamics analysis in archery sports. Through this interdisciplinary approach, we gain insights into archery mechanics and a broader understanding of human motor control, focus, and cognitive processes during discrete movements in sports.

We find professional archers exhibit movements with less dispersion in their motion dynamics, likely due to more active corrections and more fluid dynamical control compared to neophytes.

## 2. Materials and Methods

### 2.1. Participants

The research involved 14 professional archers and 14 neophytes (seven men and women in each) (Table 1). Professional archers played for Poland’s national team, while the neophytes were senior physical education students who had never shot a sports bow or any other type of bow. The participants reported having no existing upper limb injuries or balance disorders and had not undergone any major upper limb surgery. The study was conducted according to the ethical guidelines and principles of the Declaration of Helsinki. The study protocol was approved by the University Research Ethics Committee (SEK 01-09/2020). In accordance with the emphasis on *safety* by the IRB, neophytes were particularly fit individuals for whom the recommended neophyte draw weight of 30 lbs. was not difficult. Also, the spirit of the project was to allow neophytes to, primarily, safely perform the task. Therefore, we asked them to perform the bow-draw motion as they felt most comfortable. Thus, for example, we did not control for eye dominance or ask them to perform the task with full stabilization or viewfinder hardware, which would have proven to be a distraction. Similarly, to further emphasize safety, they did not release the bow string to prevent the string from stinging their bow forearm.

### 2.2. Experimental Procedure

This research was conducted by utilizing a 9-camera system (Vicon Motion Systems Ltd, Yarnton, UK) operating at 100 Hz. A total of 34 markers were placed on the subjects according to the full body Plug-In-Gait scheme (Figure 1). From these markers, we extracted the 3D location of the Center of Mass (CoM), the bow hand (left arm), and the drawing hand (right arm). The CoM was calculated as the centroid of all body parts, including both hands. Each hand position was measured via markers placed on the dorsum, just below the head of the second metacarpal. After a warm-up, professionals performed two successful shots, and neophytes performed one shot (two shots if the first one was unsuccessful) at the target located 5 m from the platform. The shot cycle (Figure 1) consisted of the following phases: A. Set-up, B. Draw, C. Aim, D. Release, and follow-through [22]. The set-up phase (stage A) finishes when the bow hand is raised to the highest point before the drawing hand moves backward. The draw phase (stage B) finishes when the string touches the face of the archer. The Aim phase (stage C) finishes when the string moves forward from the fingers (the release, stage D). The follow-through time stops when the archer first moves either arm downwards from their end position. The starting point of recording is when the drawing hand and the LASI (Left Anterior Superior Iliac Spine) are at the same height (Figure 1).

The professional archers performed well-aimed shots at the center of the target (A, B, C, and D of the shot cycle, Figure 1), while the neophytes drew the string without taking the arrow down from the bow (for safety reasons, they only performed phases A, B, and C of the shot cycle before reversing the movement back to A, Figure 1). The professional archers had their bow with a draw weight set between 36 and 40 lb. For neophytes, the draw weight was set to 30 lb as our consultant coaches recommended using bows with a draw weight of no more than 30 pounds for neophytes. Before the recorded trials, the neophytes did not learn the technique themselves, but instead watched a professional archer perform the bow draw. Our analysis for each participant was carried out on the first attempt performed without marker occlusion.

### 2.3. Data Preprocessing

#### 2.3.1. Kinematics Resampling and Normalization

Using phase space reconstruction and Hurst exponent analysis requires the signal to have a large enough number of points relative to their sampling frequency. Given our sampling frequency of 100 Hz, the lengths of the bow-draw movements lasting at most 4.2 s (for professionals) and 9.7 s (for neophytes) did not provide the same number of points per movement. Therefore, we preprocessed and concatenated the signals as follows: (I) The signals were resampled to obtain 1000 samples for the representative movement by each participant. This avoided undersampling (removing information from) the longest (i.e., slowest) movement, which had 970 samples. (II) Then, the resampled signals were normalized by the maximum magnitude value and demeaned. (III) To make the movements continuous, every other time series was reversed; therefore, the time series could be seen as a cyclical movement for the phase space reconstruction and Hurst exponent analysis. (IV) Finally, a low-pass filter was applied to prevent impulsive jumps between concatenated time series ($f_{c}=0.66$ normalized frequency).

#### 2.3.2. Creation of a 1D Time Series for the CoM, Drawing Hand, and Bow Hand

Phase space reconstruction and Hurst exponent analysis both require a 1D time series. Thus, we applied principal component analysis (PCA) to each body part’s 3D time series to reduce the dimensionality (See Figure 2). PCA is a widely employed technique in the fields of biomechanics [23,24] and sports [25,26] to project data into lower dimensions efficiently. The variance explained by the first PC of all body parts is shown in Table 2. In Appendix A, we further analyze the first PC for each body part to determine if they exhibited nonstationary, nonlinear, or chaotic behavior.

### 2.4. Phase Space Reconstruction

#### 2.4.1. Time Delay and Embedding Dimension

The modeling and visualizing of time-series dynamics require proper state space reconstruction from available data to successfully estimate the invariant properties of the embedded attractor. Therefore, an appropriate time delay (τ) and embedding dimension (*D*) should be selected for the phase space reconstruction [27]. In the case of this paper, time-delayed embedding has been used to embed a signal into higher-dimensional space. To estimate the optimal time delay value, the Average Mutual Information (AMI) method was used for the first PC of each body part to implement the uniform multivariate method [28]. The criterion used for choosing the time delay was to find the first local minima of the AMI function for all body parts in both groups, and the maximum time lag ($\tau_{max}$) was set to be 1000 samples (the length of one participant’s time series). The dimension was found using the False Nearest Neighbor (FNN) method, which computes the percentage of false nearest neighbors for multidimensional input time series as a function of the embedding dimension [29]. All three first PCs showed a dimension of three.

#### 2.4.2. Convex Hull of Phase Portrait

For the prepared data, by applying the optimal time delay (τ) and the embedding dimension (D=3), a convex hull was calculated for each PC in both groups (Figure 3). The convex hull’s volume was used to compare the phase space domains. Phase space volumes were compared within the first PC of the three body parts and between groups (at the level of the same body part).

### 2.5. Hurst Exponent Analysis

The Hurst exponent (*H*) quantifies the time series’ long-term memory by examining the auto-correlations within the series and how these correlations diminish as the time lag between data points increases [30,31]. *H* represents the relative tendency of a time series to have a mean-reverting or trending pattern. The Hurst exponent can be between 0 and 1, categorizing the series into three types: (1) H>0.5, which shows a persistent series. The closer the *H* value to 1, the stronger the persistence. (2) H=0.5, which shows a Brownian motion (i.e., random motion). (3) H<0.5, which shows an anti-persistent series. The closer the *H* value is to 0, the stronger the anti-persistence is. Taken together, H>0.5 or H<0.5 shows that a higher value comes after a high value, and vice versa [32]. To calculate the Hurst exponent, we used the Rescaled Range (R/S) Analysis [30]. We define X(t) as the first PC with the length of *N*, and X(i) is the *i*th observation in the PC. R(n) is the range of the series over a segment of length *n*, defined as (Equation (Equation 1))
(1)$$R\left(n\right)=\max_{i=1,\ldots ,N-n+1}X\left(i+n-1\right)-\min_{i=1,\ldots ,N-n+1}X\left(i\right)$$S(n) is the standard deviation of the PC over the same segment, defined as: Equation (Equation 2):(2)$$S\left(n\right)=\sqrt{\frac{1}{n-1}\sum_{i=1}^{i=n}{\left(X\left(i\right)-\mu_{n}\right)}^{2}}$$
where $\mu_{n}$ is the mean of the series over the segment of length *n*. The R/S statistic is defined as Equation (Equation 3):(3)$$R/S\left(n\right)=\frac{R(n)}{S(n)}$$The Hurst exponent *H* is then estimated using the log(R/S) versus log(n) plot as follows: Equation (Equation 4):(4)log(R/S(n))=C+Hlog(n).
where *C* is a constant and *H* is the Hurst exponent. All calculations were performed using MatLab software, version MATLAB R2023a.

### 2.6. Sample Entropy

Sample entropy (SampEn) is a modified version of Approximate Entropy (AppEn) that measures complexity or irregularity in time-series data as AppEn does, but it does not consider self-similarity. For a given embedding dimension (*m*), tolerance (*r*), and data length (length(x)=N), we define a temple vector of length *m* to be $X_{m}\left(i\right)=x_{i},x_{i+1},\ldots ,x_{i+m-1}$ and a Euclidian distance function $dij=\sqrt{\sum_{k=1}^{m}{\left(X_{m}\left(i\right)-X_{m}\left(j\right)\right)}^{2}}$, then the SampEn is as in Equation (Equation 5)
(5)$$SampEn=-\log\left(\frac{C(m,r)}{C(m+1,r)}\right)$$In which *C* is a count function as in Equation (Equation 6)
(6)$$C\left(i,r\right)=\sum_{j=1,j\neq i}^{N-(m-1)}\Theta \left(r-d_{ij}\right),\;$$
and Θ is the Heaviside step function.

Since C(m,r) is always smaller than or equal to C(m+1,r), the SampEn will always be non-negative. A smaller value of SampEn indicates more regularity (less complexity) in the data.

## 3. Results

The mean trajectories of all body parts were plotted, and the area of ±standard deviations was shaded (Figure 4). Trajectories generally reveal a difference between the two groups. To investigate dynamical differences between professionals and neophytes, phase portraits were plotted for the first PC of the concatenated time series (Figure 5).

All body parts show less dispersed dynamic states (i.e., tighter control) in professionals compared to neophytes. We used the volumes of the convex hulls to measure the dispersion for dynamic states of participants’ control during the bow-draw motion (Table 3). Based on the volumes, all three body parts have smaller dispersion in professionals, and the CoM has the greatest difference.

Hurst exponent values indicate a more persistent drawing behavior in the bow hand and CoM in professionals than neophytes (see Figure 6). A time series with 0.5<H<1 is considered persistent, and the closer to one the *H* value is, the more persistent the time series is. All of the series in this paper were persistent (Table 4), and the *H* values for the professionals were smaller in all body parts for the professionals compared to the neophytes, but bigger differences were for the bow hand and the CoM, which shows a less persistent control (i.e., more active correction) for the professionals that makes their bow draw more distinctive.

SampEn values were smaller for all body parts in the professionals compared to neophytes, with bigger differences in the drawing hand and center of the mass, indicating that professionals have a less regular drawing style.

Each population (professional or neophyte) is characterized by a single concatenated time series of its participants. To validate our results, we employed bootstrapping to generate 1000 resampled datasets from each population to estimate the underlying distribution of the analyzed signals. Bootstrapping is, therefore, a computational means to enhance the power of statistical tests by simulating data collection in many more participants with similar simulated behavior as the actual participants. We subsequently repeated our analyses and used *t*-tests to estimate the statistical robustness of differences in test statistics among the resampled groups. Remarkably, except for the bow hand’s sample entropy (i.e., 1 out of 9 test statistics), all other comparisons yielded statistically significant differences (*p*-value < 0.01), strongly suggesting that the differences in the features from the actual professional and neophyte participants are indeed real, despite the limited number of subjects.

## 4. Discussion

As expected from their skill levels, there were differences in how professional and neophyte archers executed the bow-draw motion. However, our contribution is to be able to quantify the *dynamical features* that can capture those differences for this discrete (i.e., non-cyclical) bow-draw task. We show this from a variety of perspectives that include *qualitative* between-groups differences in the normalized 3D trajectories (Figure 4), and *quantitative* differences illustrated by dynamical time-series analyses: phase portraits (Figure 5) and Hurst exponents (Figure 6 and Table 4) and sample entropy (Table 5).

It is known that postural stability is a trainable attribute that can be enhanced through consistent practice, as demonstrated in studies by Jagdhane et al. (2016) [33] and Paillard (2017) [34]. In archery, specifically, an examination of postural balance during the aiming phase has revealed that better performance in professional archers, compared to their less-skilled counterparts, can be attributed to reduced postural sway characteristics. Our findings of lower dispersion (i.e., less variability and thus reduced sway) in the movement of the center of mass and hands for the professionals compared to the neophytes agree with those findings [18,19,20,35,36]. Our results critically extend those statistically based studies by revealing that differences in the dynamics of bow-draw motion between professional and neophyte archers can be attributed to different *motor control strategies*. This is because dynamical features such as smaller state space dispersion, more active corrections (less persistence in Hurst exponent analysis), and less regularity (in sample entropy) also have control of theoretical implications and interpretations.

Moreover, this work also serves as proof of principle that such dynamical time-series analyses can be applied to discrete athletic movements. Historically, dynamical systems analysis has been applied mostly to continuous motions such as quiet stance [37], gait [38,39], postural control [40] and running [41]. This is particularly useful in sports because many sports center around discrete actions. We note this because, before this work, most of these analyses were applied to inherently cyclical sports such as walking, running, and swimming.

Our preprocessing of their time series made applying these methods to the discrete bow draw possible. In particular, we first resampled, normalized, and demeaned the signals to prevent bias due to participant anatomy and movement duration differences. We then concatenated each instance of a movement from each participant group (reversing every other movement). This created a continuous time series amenable to these analyses that represented each participant while producing a balanced and long-enough time series representative of each group. This preprocessing of the time series from individuals’ motion into a single 1D time series provides only one value for the convex hull’s volume and Hurst exponent for each group. Nevertheless, this single number represents each group and lets us compare them as a group. Comparisons of individual subjects were not our goal.

Before interpreting our results, it is important to state some particular limitations of our project. As shown in Figure 1, only the experts completed the task by releasing the arrow (stage D of the shot cycle). This was imposed for the sake of the safety of the neophytes, as the release phase can be dangerous. Therefore, neophytes only reached the Aim phase (stage C) before relaxing. However, this does not affect our analysis of the bow-draw movement as the Aim phase is the end of the same. In addition, before performing the bow draw, the neophytes’ training consisted only of watching a professional archer perform the bow draw and release in silence. Moreover, they did not receive additional instruction, and we only analyzed their first motion for which markers were least occluded. The first trial analyzed only a truly naïve shot. Offering instruction would have added to the confound of learning, which was not the goal of our study. Although we expected this lack of formal instruction could result in greater inconsistency in the neophytes than in the professionals, this did not wash out-group differences, as demonstrated by Table 3 and Table 4. In addition, the purely visual exposure to the task before their bow-draw attempts emphasized learning by demonstration in the neophytes. The consistency we found in their movements (see Table 4) may come from their perception of the most salient visual features of the bow draw, as opposed to the details of motor performance of the task. Regardless of whether the task was being imitated at a perceptual or motor level by the neophytes (i.e., we cannot espouse either at this point), we find differences in the motor performance of the task across groups as described below. Lastly, we did not study, and therefore did not test for, the difference in dynamic performance between the sexes. All participants could easily draw the 30 lb. draw weight, and the average and standard deviation in weight and height between the two groups were similar (except for a greater standard deviation in the neophyte height). Future work that is properly powered for this comparison can assess differences in motor control between the sexes in this sport.

Phase portraits are geometrical representations of the transitions in the dynamical states of a system. Therefore, the volume of their convex hull represents the breadth (dispersion) of its dynamics. Professionals exhibited phase portraits with smaller volumes, indicating that the underlying dynamics of their motion were controlled to have less dispersion—which can be interpreted as having tighter control over the range of positions, velocities, and accelerations of their body parts.

A Hurst exponent of 0.5 represents a truly random Brownian motion. A value closer to 1.0 then quantifies ‘persistence’ in a time series (i.e., the tendency to continue the current trend of a movement). The Hurst exponents we found show that professionals are less persistent as a group (closer to 0.5) compared to the neophytes (closer to 1.0). This can be interpreted as professionals implementing more frequent and minute corrections, resulting in shorter movements.

This study utilizes Sample Entropy (SampEn), a modified adaptation of Approximate Entropy, as a quantitative metric for gauging complexity and irregularity within time series while circumventing the influence of self-similarity inherent to Approximate Entropy. Higher SampEn values indicate more complexity, signifying lower regularity within the time series. Our comprehensive analysis consistently reveals that professional archers exhibit higher sample entropy values across all three body parts under scrutiny, indicating more complexity than neophyte archers. The higher sample entropy values observed for the professional archers can manifest from a heightened adeptness in orchestrating a fluidic and precisely directed bow-draw motion. We conclude that professional archers exhibit tighter and finer control over their discrete bow-draw movements’ more fluid (i.e., less regular) dynamics. Although differences between these groups were expected, our work provides proof of principle of how well-established dynamical analyses can be used to quantify and compare the dynamics of discrete movements in sports.

## Figures and Tables

**Figure 1 entropy-25-01414-f001:**
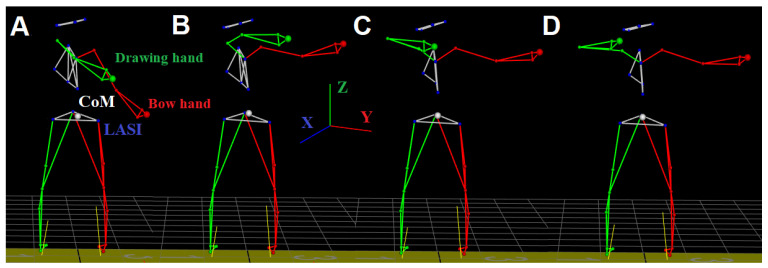
Motion capture stick figure of a professional participant during the shot cycle: (**A**) Set-up phase, (**B**) Draw phase, (**C**) Aim phase, (**D**) Release phase (not included in the analysis and not done by neophytes).

**Figure 2 entropy-25-01414-f002:**
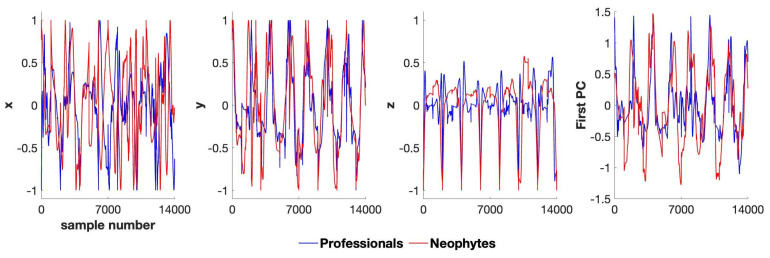
Preprocessed and normalized time series (of each group) of the Cartesian coordinates of the center of mass (CoM) and their projection onto their first principal component (PC). We resampled the bow-draw movement from all participants (which could have taken a different amount of time) to have 1000 samples each and then concatenated them to create the time series for each group. Considering the number of participants in each group (N = 14), the concatenated time series has 14,000 samples for neophytes and professionals.

**Figure 3 entropy-25-01414-f003:**
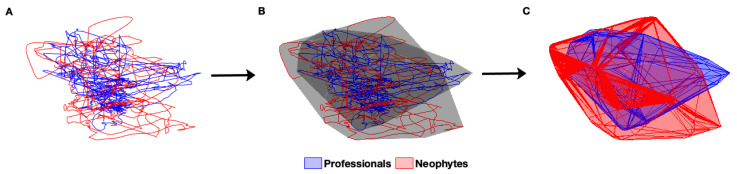
Steps for: (**A**) Plotting the phase portrait for the 1D time series of the CoM using the time-lag method, (**B**) Fitting the convex hull to the phase space trajectories, and (**C**) Removing the trajectories from the convex hulls to generate Figure 5c.

**Figure 4 entropy-25-01414-f004:**
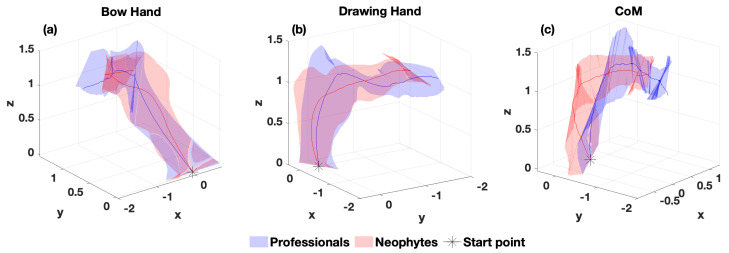
Normalized Cartesian coordinates of the hands (**a**,**b**) and CoM (**c**) for the single bow-draw motion for all participants. The asterisks indicate the start of the motion, and the shaded areas represent standard deviations from the mean presented by the solid line.

**Figure 5 entropy-25-01414-f005:**
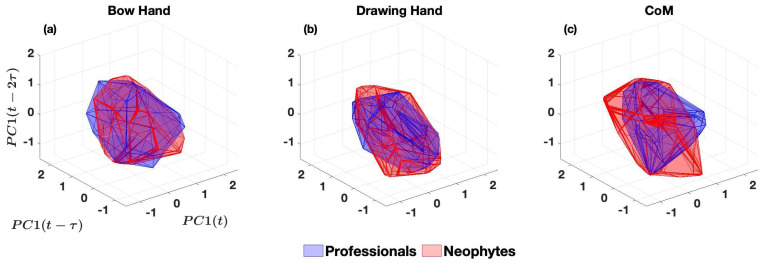
Convex-hulls of the phase portraits for the bow hand (**a**), the drawing hand (**b**), and CoM (**c**) of the professionals and neophytes. There is one convex hull per group, as each phase portrait was obtained from the concatenated time series from all subjects in that group. The volume of each convex hull is shown in Table 3.

**Figure 6 entropy-25-01414-f006:**
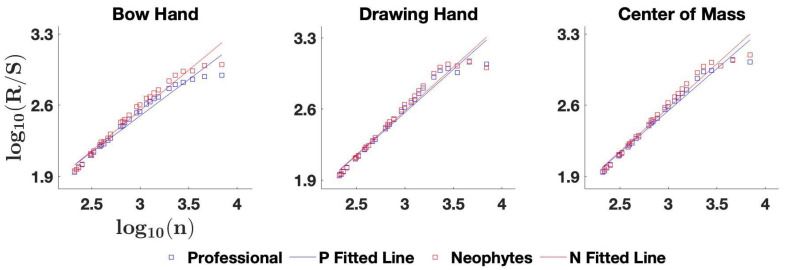
The slope of the fitted line on the calculated data points from the logarithm of rescaled range (R/S) vs. the logarithm of time lags is equal to the Hurst exponent.

**Table 1 entropy-25-01414-t001:** Participant characteristics (mean ± SD).

Group	Age [Years]	Body Mass [kg]	Height [cm]	Training History [Years]	Draw Weight [lb]
Professionals (*N* = 14)	23.7 ± 9.9	73.2 ± 16.5	175.7 ± 9.7	11.1 ± 7.9	36–40
Neophytes (*N* = 14)	23.5 ± 1.3	73.3 ± 16.7	176 ± 13.5	0	30

**Table 2 entropy-25-01414-t002:** Percentage of variance explained by the first PC for each body part.

Body Part	Professionals	Neophytes
Bow Hand	64.3	68.6
Drawing Hand	76.1	74.9
Center of Mass	56.5	62.5

**Table 3 entropy-25-01414-t003:** Convex hull volumes (normalized units) for the phase portraits of each body part. * denotes statistical significance with p<0.01.

Body Part	Professionals		Neophytes
Bow Hand	7.0	< *	8.3
Drawing Hand	7.3	< *	8.0
Center of Mass	6.3	< *	11.2

**Table 4 entropy-25-01414-t004:** Hurst exponent (*H*) values for each body part. * denotes statistical significance with p<0.01.

Body Part	Professionals		Neophytes
Bow Hand	0.71	< *	0.79
Drawing Hand	0.85	< *	0.86
Center of Mass	0.82	< *	0.86

**Table 5 entropy-25-01414-t005:** Sample Entropy values for each body part. * denotes statistical significance with p<0.01.

Body Part	Professionals		Neophytes
Bow Hand	0.024	>	0.023
Drawing Hand	0.020	> *	0.014
Center of Mass	0.035	> *	0.025

## Data Availability

The measurement data used to support the findings of this study are available from the second author upon request.

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
