# Peer review of "Dynamical Analyses Show That Professional Archers Exhibit Tighter, Finer and More Fluid Dynamical Control Than Neophytes"

_entropy, 2023, doi:10.3390/e25101414_

Round 1
Reviewer 1 Report
Thank you for the opportunity to review the work. Its an interesting piece of work which is an excellent representation of methodology application to sport.
Some small points to expand on/clarify
Ln49 - please be clear on how many shots each participant took
Section 2.1 - is there a gender split at all? Could any of that affect or explain some of the results? If this isnt important then please explain why in text (somewhere).
Ln81 - give details of the low pass filter.
Figure 2 - needs editing to match the 1000 samples discussed in section 2.3.1
Ln 112: please rephrase this, the discussion of a random walk doesn't make sense, please edit the language here
In the discussion it could be developed with wider application to general sport - how is this useful beyond methodological / mathematical understanding? How can we use it in practice?
Author Response
Please see the attachment for the reviewer 1. Thanks.

Reviewer 2 Report
The authors present an original and very interesting paper investigating differences between professional archers and neophytes related to dynamic control during shooting movement and its phases.
This is an area that has received a little attention in the literature, therefore, warrants further examination. Overall, the manuscript is written and organized fairly well. It follows the logical sequence of a research purpose. Despite this strength, I have few comments that need to be addressed by the authors and listed below.
INTRODUCTION
This section should be improved. Indeed, a single paragraph only related to the dynamic systems theory (tools and techniques investigating athletic performance) is incomplete to really present the aim of the study.
Even if phases of the shooting movement are presented in Materials and Methods, authors should introduce why archery has been chosen for the purpose of their study. Studies presenting the shooting movement of archers (neuromuscular control and coordination, postural control and learning) are missing.
Some are listed below and should be used too in the discussion (see comment in discussion section):
Dorshorst T, Weir G, Hamill J, Holt B. Archery's signature: an electromyographic analysis of the upper limb. Evol Hum Sci. 2022 May 26;4:e25. doi: 10.1017/ehs.2022.20.
Vendrame E, Belluscio V, Truppa L, Rum L, Lazich A, Bergamini E, Mannini A. Performance assessment in archery: a systematic review. Sports Biomech. 2022 Mar 29:1-23. doi: 10.1080/14763141.2022.2049357.
Baifa Z, Xinglong Z, Dongmei L. Muscle coordination during archery shooting: A comparison of archers with different skill levels. Eur J Sport Sci. 2023 Jan;23(1):54-61. doi: 10.1080/17461391.2021.2014573.
Sarro KJ, Viana TC, De Barros RML. Relationship between bow stability and postural control in recurve archery. Eur J Sport Sci. 2021 Apr;21(4):515-520. doi: 10.1080/17461391.2020.1754471.
Soylu AR, Ertan H, Korkusuz F. Archery performance level and repeatability of event-related EMG. Hum Mov Sci. 2006 Dec;25(6):767-74. doi: 10.1016/j.humov.2006.05.002.
Shinohara H, Urabe Y. Analysis of muscular activity in archery: a comparison of skill level. J Sports Med Phys Fitness. 2018 Dec;58(12):1752-1758. doi: 10.23736/S0022-4707.17.07826-4.
Callaway AJ, Wiedlack J, Heller M. Identification of temporal factors related to shot performance for indoor recurve archery. J Sports Sci. 2017 Jun;35(12):1142-1147. doi: 10.1080/02640414.2016.1211730.
Beyaz O, Eyraud V, Demirhan G, Akpinar S, Przybyla A. Effects of Short-Term Novice Archery Training on Reaching Movement Performance and Interlimb Asymmetries. J Mot Behav. 2023 Aug 16:1-13. doi: 10.1080/00222895.2023.2245352.
Spratford W, Campbell R. Postural stability, clicker reaction time and bow draw force predict performance in elite recurve archery. Eur J Sport Sci. 2017 Jun;17(5):539-545. doi: 10.1080/17461391.2017.1285963.
Ertan H, Kentel B, Tümer ST, Korkusuz F. Activation patterns in forearm muscles during archery shooting. Hum Mov Sci. 2003 Feb;22(1):37-45. doi: 10.1016/s0167-9457(02)00176-8.
Laborde S, Dosseville FE, Leconte P, Margas N. Interaction of hand preference with eye dominance on accuracy in archery. Percept Mot Skills. 2009 Apr;108(2):558-64. doi: 10.2466/PMS.108.2.558-564.
Kuch A, Tisserand R, Durand F, Monnet T, Debril JF. Postural adjustments preceding string release in trained archers. J Sports Sci. 2023 Jun;41(7):677-685. doi: 10.1080/02640414.2023.2235154.
MATERIALS AND METHODS
2,2 experimental procedures. For neophytes, authors should stipulate if eye dominance has been determined. Authors should justified why 30lbs has been chosen. For some neophytes this draw weight could have been too high. Finally, from an ecological point of view, why authors have decided to not observe all participant with the same material, only a bow (without the system of stabilization and viewfinder)?
I understand that neophytes only viewed the movement to realize by watching professional archers to ensure that there is no learning effect due to practice. How did the authors prevent one of the most frequent injury observed with neophyte in archery: the string that stings the forearm when there is not sufficient internal rotation of the bow arm shoulder to lock the joint and ensure safety?
Statistical analyses.
There is no statistical analyses presented to compare and observe significant differences between professional archers and neophytes for manipulated variables. A section should be added and results section should integrate these statistics for the presentation of the “observed” differences.
RESULTS
This section should be improved. See previous comment.
DISCUSSION
The discussion is very interesting and well documented. However, even if I understand that the discussion is mainly focused on the reliability of dynamical time series analyses for discrete athletic movements such as archery, authors should link and discuss their results too with current literature and results that observed shooting movement and specific variables quantifying this particular movement and its phases.
Author Response
Please see the attachment for the reviewer 2. Thanks.

Round 2
Reviewer 2 Report
Thank you for your work and for all improvements made in the revised version of your manuscript. It has been a real pleasure to evaluate this work (espescially for a former member of the French archery team).